# The Causality and Uncertainty of the COVID-19 Pandemic to Bursa Malaysia Financial Services Index’s Constituents

**DOI:** 10.3390/e24081100

**Published:** 2022-08-10

**Authors:** Daeng Ahmad Zuhri Zuhud, Muhammad Hasannudin Musa, Munira Ismail, Hafizah Bahaludin, Fatimah Abdul Razak

**Affiliations:** 1Department of Mathematical Sciences, Faculty of Science & Technology, Universiti Kebangsaan Malaysia, Bangi 43600, Selangor, Malaysia; 2Department of Computational and Theoretical Sciences, Kulliyyah of Science, International Islamic University Malaysia, Kuantan 25200, Pahang, Malaysia

**Keywords:** transfer entropy, granger, bursa Malaysia financial services, COVID-19, large-cap, mid-cap, small-cap, insurance, directed causal temporal networks

## Abstract

Valued in hundreds of billions of Malaysian ringgit, the Bursa Malaysia Financial Services Index’s constituents comprise several of the strongest performing financial constituents in Bursa Malaysia’s Main Market. Although these constituents persistently reside mostly within the large market capitalization (cap), the existence of the individual constituent’s causal influence or intensity relative to each other’s performance during uncertain or even certain times is unknown. Thus, the key purpose of this paper is to identify and analyze the individual constituent’s causal intensity, from early 2018 (pre-COVID-19) to the end of the year 2021 (post-COVID-19) using Granger causality and Schreiber transfer entropy. Furthermore, network science is used to measure and visualize the fluctuating causal degree of the source and the effected constituents. The results show that both the Granger causality and Schreiber transfer entropy networks detected patterns of increasing causality from pre- to post-COVID-19 but with differing causal intensities. Unexpectedly, both networks showed that the small- and mid-caps had high causal intensity during and after COVID-19. Using Bursa Malaysia’s sub-sector for further analysis, the Insurance sub-sector rapidly increased in causality as the year progressed, making it one of the index’s largest sources of causality. Even after removing large amounts of weak causal intensities, Schreiber transfer entropy was still able to detect higher amounts of causal sources from the Insurance sub-sector, whilst Granger causal sources declined rapidly post-COVID-19. The method of using directed temporal networks for the visualization of temporal causal sources is demonstrated to be a powerful approach that can aid in investment decision making.

## 1. Introduction

The rigor of a corporation raising capital through an initial public offering (IPO) process, regulated by the Securities Commission Malaysia [1] (SC) and its subsequent listing managed by the exchange holding company, Bursa Malaysia Berhad (Bursa Malaysia) [2,3], can be taxing and expensive [4,5,6,7,8]. The corporation’s progression on three key Bursa Malaysia markets [9] starting from the emerging market, to the growth-based market and, finally, to the established market can be demanding as well. These three key markets are known as the Leading Entrepreneur Accelerator Platform (LEAP) Market; Access, Certainty and Efficiency (ACE) Market; and the Main Market. Regardless of the challenges, if the public sentiment favors the corporation’s prospects, the endeavor of an IPO could be strategically and financially rewarding [6]. 

At the corporation’s peak, it may even reach the zenith of the Main Market index, which is the Kuala Lumpur Composite Index (KLCI) reflecting the largest outstanding share or market capitalization (cap) valued in hundreds of billions of ringgit. The KLCI’s 30 highest performing constituents are also known as the blue-chip stocks [10,11] consisting of many of the large-cap [12] corporations. These large-cap constituents make up 30% of the Bursa Malaysia top 100 best performing constituents.

Nevertheless, regardless of the corporation’s ability to reach Bursa Malaysia’s zenith, residing in the mid-range or even in the smaller market cap, may show signs of growth or resilience during difficult periods, such as the most recent spread of the highly contagious or even deadly Coronavirus Disease 2019 (COVID-19) [13]. Conversely, an underperformance may not only discourage investors but also influence the perception of other listed companies, casting a cloud of negative press on the overall market’s performance. 

Therefore, it is pertinent for the survival of the listed corporations to maintain their performance [8,14]. According to [15], 86.7% Malaysian listed corporations survived 7 years after their initial IPO. With this insight, Bursa Malaysia’s indices and its data are of high value to investors as it is an indicator of the corporation’s performance, yielding business intelligence to the investors. Hence, this is one of the many reasons that the market indices can be seen as one of the many barometers for the country’s economic health [16,17].

### 1.1. The Main Market Sectorial Indices and the KLFIN

Within Bursa Malaysia’s Main Market, 13 further sectorial indices (sectors) are available, including construction; consumer products and services; energy; financial services; health care; industrial products and services; plantation; property; real estate investment trusts; technology; telecommunications and media; transportation and logistics; and utilities [18], with varying amounts of outstanding shares.

Looking through the recent KLCI performance on the Russell Group’s Financial Times Stock Exchange report [19] and the latest Bursa Malaysia’s report on sectorial index series [18], the financial services sector dominates the Main Market. In 2021 alone, its total market capitalization was the highest at RM 362.02 billion; the closest according to the same report [18] was Consumer, Product and Services at RM 260.07 billion. This naturally cements the financial services sector’s dominance in Bursa Malaysia even post-COVID-19.

Bursa Malaysia’s index that tracks the financial sector is known as the Bursa Malaysia Financial Services Index (BM Financial Services) or sometimes referred to as the Kuala Lumpur Finance Index [20,21,22,23,24], the Kuala Lumpur Financial Index [25] (KLFIN) or the BM Financial Index [26]. While it is clear that the index has varying designations from the cited works, for practical abbreviation purposes, this paper refers to the index as the KLFIN, as more cited works from the academic literature [22,27], industry reports [20,25] and even Bursa Malaysia [21] use this abbreviation. Within this index is also several Shariah-compliant constituents where business activities are governed by Islamic law [28,29,30,31,32,33,34].

### 1.2. Motivation and Impact behind Causal Analysis Using the KLFIN

Since it is known that the financial sector is the strongest and has several large-cap constituents within the index, a question is raised: is there any causal effect within the KLFIN’s individual constituent regardless of its market size? Would the individual constituent’s sub-sectors within the KLFIN have any impact against each other? If so, to what extent, and can it be measured? Although it is known that the KLFIN is the most lucrative segment within the Main Market, it is however reasonable to postulate that the KLFIN will not be immune to harsh or uncertain times; thus, the performance of the individual constituents may causally influence each other. 

Thus, this paper hypothesizes that, during and after COVID-19, the causal intensity of these lucrative individual constituents will intensify regardless of their prominent position in Bursa Malaysia. The novelty of knowing the causal behavior of the individual constituents, regardless of their stature, during uncertain times, such as COVID-19, can uncover novel perspectives of the constituent’s influential power during such extreme times. 

Furthermore, having a numerical measure of causal effect would be of value to investors and analysts as Bursa Malaysia’s KLFIN is no longer seen as simply a list of market benchmark [17] but an extended insight for measuring the causal effects between constituents regardless of its position in Bursa Malaysia or business ventures. As far as this paper is aware, no such investigation at the level of the constituents has been conducted in Malaysia.

Since the keyword is temporal financial data and its causality, it is fitting that the first objective of this paper is to use the Nobel laureate Clive Granger’s famous econometrics work on causality [35]. Subsequently, the second objective of this paper is to use Schreiber transfer entropy as an alternative causality measure but in a non-linear sense. Finally, for the third objective, since numerical measures are of interest, this paper extends Granger and Schreiber’s causal statistical and mathematical approaches using network science, thus, allowing the extended power of measuring the fluctuating source of causality in a network form.

The results showed that both networks were successful in not only capturing sources of causal extremes but also, interestingly, the findings indicated that both networks overlapped in their results even though both differed in their implementation. Unexpectedly, both networks showed that the small- and mid-caps had high sources of causal intensity during and after COVID-19. As both networks were able to detect causal extremes even in the KLFIN with differing intensities, the hypothesis that, during and after COVID-19, the causal intensity of these lucrative individual constituents will intensify regardless of its prominent position in Bursa Malaysia is accepted.

The rest of the paper is organized as follows. Section 2 presents the literature review on the large-cap and its relativeness to the other market caps, as well as the literature for Bursa Malaysia Financial Services Index, Granger causality and Schreiber transfer entropy. Section 3 illustrates the methods used. Section 4 presents the analysis done by this paper. Section 5 presents the discussions on the findings. Finally, Section 6 presents the conclusion of the paper.

## 2. Literature Review

Within a specific index, the large-cap (sometimes referred to as the big-cap) can peak towards hundreds of billions of ringgit in outstanding stocks; this can be seen as stable and reliable [36] and, thus, naturally attractive. Based on Bursa Malaysia, the market capitalization threshold for the mid-cap is from 200 million to less than 2 billion, and the small-cap is less than 200 million [37]. Note that this definition may differ based on the sources [38,39,40]; for standardization, this paper follows [37]. Nevertheless, regardless of the definition, would any of these caps be favorable during uncertain times, such as COVID-19?

This section explores the literature and financial reports from the relevant sources describing the caps’ performances during the pre- and post-COVID-19. The next sub-section will explore the performance of the caps and, afterwards, the definition and applications of causality.

### 2.1. Bursa Malaysia’s Large-, Mid- and Small-Cap: Performance Comparisons Pre- and Post-COVID-19

#### 2.1.1. Performance of the Caps

This sub-section starts with the understanding of the perceived stability and reliability of the large-cap. Afterwards, it explores the appreciation of the stock price value within a specific cap (e.g., mid- or small-cap) as well as its demands within a decade, comparatively to the large-cap.

Based on the most recent reports from Russell Group’s Financial Times Stock Exchange [19], RHB (using Bloomberg data) [41] and the SC’s Capital Market Masterplan 3 [42], the prior question of stability and reliability of the large-cap seemed to be untrue, as the large-cap is also subjected to underperformance. The most critical piece coming from RHB’s report was the following statement, “favoring the large-cap during a down period does not hold any ground”. The report indicates that investors will seek other caps if growth can be found elsewhere. Moreover, RHB also reported that, despite unfavorable and uncertain times for the country and the world, it was the small- and mid-caps that performed during a bullish period in 2020, i.e., post-COVID-19.

Further to RHB’s analysis [41], the cited bank also illustrates the yearly returns (i.e., the appreciation of the price) of the stocks within each cap over a decade; surprisingly, it was the FBMSC (i.e., small-cap) that gained multiple times during COVID-19. This pales in comparison compared to the blue-chip (i.e., FBMKLCI), which was in the negative, and the FBM70 (i.e., capitalization smaller than the FBMKLCI but larger than FBMSC), which was almost 3× lower than the small-cap.

Looking at the turnover of the caps (i.e., demands for the stock in a particular cap), this paper refers to the Bursa Malaysia MidS index, launched in 2017 [42], which tracks the growth of the small- and mid-cap performance. Based on the cited SC’s analysis as well, the MidS index grew at 5.5% from 2011–2020. The SC also stated that it was the small-cap that grew whilst the mid- and large-cap went downward. After 2019, the small- and mid-caps grew rapidly compared to the large-cap. This growth may be explained by the SC’s report; lower trading costs (with a stamp duty waiver) and greater liquidity would have induced growth.

The waivers offered by the SC [3], primarily benefitting the small- and mid-cap, could reduce fundraising costs and ensure further financial relief to the applicable constituents. It appears that, in 2020, the MidS fared better than the large-cap [43].

#### 2.1.2. The LEAP and the ACE Markets: Inducements of Growth to the Mid- and Small-Caps

The IPO growth and trading activities on the small- and mid-caps were also enabled by the inducement of the LEAP and the ACE Markets [42], and the LEAP Market is Bursa Malaysia’s alternative capital raising platform for small- and medium-size enterprises (SME). This helps raising the SME’s visibility on the capital market regulated by the SC [44]. Progression-wise, the ACE Market is a platform that prepares the LEAP Market to transition to the Main Market [3]. The LEAP and the ACE Markets can be attractive to SMEs (and could encourage them to be ranked as a part of the Main Market) due to their more relaxed requirements to be a part of these markets [45]. The large-cap does not have such luxury with regard to relaxed requirements [5].

While it appears that the MidS could outperform the large-cap with the more relaxed requirements, this is a superficial view, and the point of view of stability needs to be reoriented; one needs to consider that post IPOs may not even make it through the LEAP or the ACE Market; furthermore, it has been reported that constituents on the ACE have a higher probability of being delisted from Bursa Malaysia [15]. Accounting for some undesirable news, such as financial fraud and civil suits, which is seen more on the ACE Market compared to the Main Market, may discourage investors from investing with the ACE Market [46]. However, as a concluding remark, it is interesting to note that, on the ACE Market, the survival rate of the financial sector being listed under ACE is high [15]. Thus, the expansion of the Main Market is not relative to the sheer strength of the size of the market cap alone. 

As one can tell from this sub-section, being in the large-cap does not promise constant growth, and being in the MidS does not mean that it could pull the investor into the abyss of non-growth territory. The next section will look into the index that monitors the performance of the financial sector. Several academic and industrial studies and reports are presented.

### 2.2. Bursa Malaysia Financial Services Index: Perception and Evolution

Although the financial sector has the highest outstanding stocks, it is not immune to external factors and losses; this section explores previous studies performed on the KLFIN, findings on the performance from both academic and industrial reports and the perception of its performance relative to its market size.

#### 2.2.1. Analysis and Performance

As of 2022 [47], there are at least 30 financial constituents on the KLFIN. The KLFIN monitors not only financial constituents that are in KLCI but also other market sizes, such as for the small- and mid-caps.

Several academic studies have been conducted on the KLFIN: ref. [48] discovery of Granger bi-directional causality between the KLFIN and the Indonesian Financial services (JKFINA) during the period 2015–2017; ref. [22] analysis of the KLFIN’s performance relativeness to other Bursa Malaysia sectors; and ref. [27] indications that the KLFIN is considered one of the top three risky sectors and the negative impact of foreign financial crisis, e.g., the Lehman Brothers’ bankruptcy to the KLFIN [49].

With regard to the factors on the KLFIN volatility or risks, the academic literature tallies with the industrial financial reports. According to the most recent Maybank stock market reports, the KLFIN rebounded with mixed results and was impacted by international factors (e.g., US markets). However, the soaring pressure of increase interest rates rebounded the interest of buying banking portfolios [25].

As mentioned, although the financial sector has the highest outstanding stocks, it is not immune to losses or external factors; it was reported by [50] that negative returns were seen during the COVID-19 period, and it was the KLFIN that went on a steeper downward trend, while the KLCI did not suffer as much during the COVID-19 period [50]. The KLFIN was also underperforming against the KLCI due to another external factor, which is the China–United States trade war; another was decreasing asset quality, i.e., increased non-performing loans.

With this in mind, the next sub-section will look at the evolution of the KLFIN constituents.

#### 2.2.2. Evolution of the KLFIN’s Total Constitution

Based on the prior cited works from both academic and industries, the KLFIN index grows or shrinks over time depending on the constituents’ performance as with any other indice in Bursa Malaysia.

Figure 1, which is illustrated using data retrieved from the Refinitiv Datastream [47], indicates that the KLFIN’s total listed constituents went down in an almost linear decent and, surprisingly, stagnated during the pre- and post-COVID-19 pandemic.

In Figure 2, zooming in between 2018 and 2021 where the pre- and post-COVID-19 occurred, the financial constituents that were a part of the KLCI did not change drastically; it was AMMB.KL that left the KLCI in 2021.

Referring to Figure 3, few of the KLFIN constituents left its respective indices. Based on the prior cited literature, industrial reports and data extracted (from Refinitiv Datastream [47]), it seems that there is no supporting evidence that the large-cap will constantly outperform the other caps.

Alternatively, turning to the measure of the degree of the various constituents’ movements (i.e., performance) relative to each other, normally performed using Pearson’s correlation analysis, up to measuring the degree of the proximity of the constituents’ performance, has been conducted by Mantegna’s foundational work on financial networks [51]. However, Mantegna’s original work revolved around Pearson’s correlation model and not a causal model. Indeed, instead of using only measures of similar movements, identifying the causality of the individual constituents to each other may provide an alternative insight to the index itself, regardless of the constituent’s stature. Extending this causal measure using Mantegna’s original financial networks may even bring in more alternative insights as well. 

However, before one can construct a causal network, causality itself needs to be defined. The next section elaborates the fundamental works on quantitative causality—in particular Granger causality—and, afterwards, the non-linear Schreiber transfer entropy.

### 2.3. Causality

Before diving into network science where a diverse measure of causality can be extended (e.g., causality-based network topology to the centrality of individual constituent’s causality), this section will look into the meaning of ‘causality’ first.

#### 2.3.1. Causality: Overview, Background and Granger Causality

Defining ‘causality’ is non-trivial [52]; however, it is pertinent to understand what it is precisely; if a universal precise definition existed, the causality of any universal phenomenon could not only be explained easily but modelled effectively. As far as this paper is aware, a universal precise definition does not exist [53]. The argument of the notion of ‘causality’ or even the rejection of its applications in mathematics and physics [54] naturally predates many of the 21st century findings [55]. Hence, in this paper, a more pragmatic or refined definition is needed.

‘Causality’, taking a more current layman’s definition [56], is the relationship between something that happens and the reason for it happening. The simplest yet easy to understand explanation is ‘flow among any phenomena’ [54,57]. The first aforementioned layman definition on this paragraph is a modern English definition and, thus, not necessarily easy to quantify. Statistically, however, this seems to be possible; Granger [35] devised a practical statistical mechanism [58] for measuring causality. Granger stated that some temporal ordering normally seen in time series data (e.g., the stock price) may infer some causal predictive ability [59]. 

In other words, a vector time-series, say $x_{t}$, is causal to another vector time-series, say $y_{t}$ if past data (or lagged values) of $x_{t}$ helped in reducing the variance of $x_{t}’s$ statistical predictability to $y_{t}$ [52,60]. Granger then placed this idea in the context of autoregressive model (AR), capturing linear features of time series data [52]; [58] can be referred to for a good description of the statistical equations applied on Granger’s view of causality as well as its application to network science (e.g., graph weights), referenced here (1):(1)$$y_{t}=\overset{N}{\underset{i=1}{\sum }}a_{11,i}y_{t-i}\;+\overset{N}{\underset{i=1}{\sum }}a_{12,i}x_{t-i}+E_{t}\left(y\right)$$

From Equation (1), N is the number of historical data, $a_{11,i}$ and $a_{12,i}\;$ are the model’s coefficients, and $E_{t}\left(y\right)$ is the prediction errors or residuals of $y_{t}$. The null hypothesis, i.e., $x_{t}$ is not causal to $y_{t}$ is accepted by utilising the *F*-test if $a_{12,i}$ = 0. It is interesting that [58] also utilized the *F*-test’s *p*-value to denote the intensity of the causality, which was followed by the recent work of [61].

Coming back to Granger’s view of causality: using some statistical predictability of some lagged values of $\;x_{t}\;$ or $y_{t}\;$ and inferring some statistical significance leading to the conclusion of causality occurring, may not be seen as an agreed description of ‘causality’ [53]. Thus it can be seen by some that Granger causality is not true causality; alternate views of Granger causality are ‘conditional dependence’ [60], ‘statistical causality’ [62], ‘predictive causality’ [57] or simply ‘$x_{t}$ contains useful information for predicting $y_{t}$’ [63]. 

It may not be a surprise that Granger’s view of causality is controversial [60]. Granger himself said during his Nobel laureate lecture [64] that he had no idea that several sources felt strongly on the topic of causality—in particular, philosophers; Granger argued that, if either timeseries variable can improve the prediction of the other, causality occurs. 

To Granger’s own surprise [64], specific terms were created for Granger’s view of causality, including ‘Granger causality’ [59], ‘Granger cause’ [53] and ‘G-cause(s)’ [65]. Any of these terms can be used if Granger’s condition for causality and the subsequent cause-effect is seen. For this work, this paper uses ‘causality’ to refer to Granger causality. It must be noted that this model is also known as Wiener–Granger Causality (WGC) or simply G-causality, as it was Wiener who introduced the notion that a variable could be causal if its predictive ability to predict a second variable is improved by including the information from the first variable [52,66]. Granger noted Weiner’s influenced to Granger causality on his Nobel laureate lecture [64]. This paper refers to Granger causality as G-causality when referring to a noun and G-causes when referring to a verb.

Nevertheless, an alternative model that could measure causality exists; the next sub-section will look into Schreiber transfer entropy.

#### 2.3.2. Causality: A Non-Linear View and the Uncertainty of the Constituents’ Performance

Unlike G-causality, which is derived from econometrics [66,67] and statistics [58], transfer entropy is derived from information theory, rooted from applied mathematics where it measures the transmission of information over a noisy channel [68]. Thus, a stark difference between the two emerged. Another stark difference is that G-causality is also known to be the best for linear models [68] but the worst for non-linear data, whilst transfer entropy works for either linear or non-linear models [69]. 

It must be noted that non-linear extensions of G-causality can be computationally expensive [66]; however, according to Granger, non-linear models reflect a more proper approach to model practical problems, which are inherently non-linear [70]. Since Schreiber transfer entropy is a non-linear approach, it can be used to measure ‘causality’ using non-linear data [71]. In this sub-section, this paper investigates non-linear Schreiber transfer entropy and its relationship with G-causality for the KLFIN.

Before transfer entropy can be compared with G-causality, the fundamentals of Shannon’s entropy, which is the basis of transfer entropy, need to be investigated first [72]. It must be noted that the understanding of information uncertainty requires similar efforts as with the understanding of G-causality; thus, the understanding of both concepts is pertinent before a comparison can be made.

Entropy is measuring any information uncertainty, diversity or randomness [71]. Low entropy implies that information uncertainty is low. Although this definition is broad, this paper will take a more practical example on ‘uncertainty’; in a capital market, uncertainty can be seen as synonymous with the term, ‘volatility’, which is, of course, more intuitive to the analysis of the stock market. The higher the volatility of the constituent, the higher the uncertainty, and thus more randomness can be seen in the constituents’ ‘information’—or in this case, its performance.

Mathematically, the prior example can be illustrated as such; assume a series of a constituent’s price performance (e.g., closing) from a particular timeseries represented as a discrete random variable X; X can be in some probability event, denoted as P. If X is in some possible state, represented as x∈ *X* where *X* is the state space or all possible representations of x, the probability distribution of X occurring in state x is donated as $P\left(X=x\right)=p_{X}$(*x*).

Therefore, the input parameter for function H representing the entropy of this discrete random is not a singular value but a series of probability distributions. Thus, the entropy of $H\left(X\right)$ can be expressed as
(2)$$H\left(X\right)=-\underset{x\;\in \;X}{\sum }p_{X}\left(x\right)\log\;p_{X}\left(x\right)$$

Commonly, entropy is expressed in base 2; thus, onward, the log indicates base 2. With lower value of $H\left(X\right)$, the certainty of the constituent’s performance is seen [73]. Moreover, if two or more constituents’ entropies are being measured, joint entropy or Equation (3) can be used.
(3)$$H\left(X,Y\right)=-\underset{x\;\in \;X}{\sum }\underset{y\;\in \;Y}{\sum }p_{XY}\left(x,\;y\right)\log\;p_{XY}\left(x,\;y\right)$$

Unlike Equation (2), joint entropy calculates the uncertainty of both constituents; $p_{XY}\left(x,\;y\right)$ denotes the joint probability of X and Y. If both constituents are statistically independent [74], Equation (3) can be written as Equation (4).
(4)$$H\left(X,\;Y\right)=H\left(X\right)+H\left(Y\right)\;$$

The joint entropy can also be expressed as the conditional entropy [54] where the uncertainties of X are calculated, given that Y uncertainties are known.
(5)$$H\left(X,\;Y\right)=\;H\left(X\;|\;Y\right)+H\left(Y\right)$$
where $H\left(X\;|\;Y\right)$:(6)$$H\left(X\;|\;Y\right)=-\underset{x\;\in \;X}{\sum }\underset{y\;\in \;Y}{\sum }p_{XY}\left(x,\;y\right)\log\;p_{X|Y}\left(x|y\right)\;$$

As of now, only the entropies of the constituents are measured; mutually shared uncertainty of these constituents can also be calculated by knowing the intersecting or shared uncertainties of the constituents. This is known as mutual information or I.
(7)$$I\left(X,\;Y\right)=-\underset{x\;\in \;X}{\sum }\underset{y\;\in \;Y}{\sum }p_{XY}\left(x,\;y\right)\log\frac{p_{XY}\left(x,\;y\right)}{p_{X}\left(x\right)\;p_{Y}\left(y\right)}$$

Thus, if I is 0, the two constituents are essentially independent of each other [75]; the higher the I, the more mutual information that is shared between the two constituents [76]. An alternate interpretation for Equation (7) can be seen as the entropy mutually shared between two entropies [54], statistical dependence between the two entropies [77] or the overlaps between entropies [78]. The relationship between Equations (2)–(7) can be seen in Figure 4.

Unless an ad hoc mechanism is introduced, e.g., time-lag [78] on either of the mutual information random variables, the similarity for Equations (2)–(7) is that none of the equations provided a form of directionality [78,79]. However, it must be noted that mutual information can be used to measure dependency [54] if $I\left(X,\;Y\right)=0$, if and only if X is independent of Y. Similar to correlation, since these equations lack directionality, it cannot be used to measure information flow [80]. On a final note, mutual information has a symmetric relationship, i.e., each pair will say the same thing about each other [81]; this is clear from Figure 4 as the two variables are basically mutually sharing information.

Apart from calculating the information gain between uncertainties, the flow of information or ‘causality’ between the two constituents is possible. Mutual information, as stated, is symmetric, and thus no sense of direction can be inferred. However, transfer entropy on the other hand is asymmetric; thus, the flow of information from a pair of constituents can be identified [78]. This can be performed by inducing temporal properties to the prior entropy equations. 

The transfer entropy notation of $T_{Y\;\to X}$ measures the information flow from Y to X or a pair of constituents; if the transfer entropy value is positive for Y to X, Y is seen as reducing the degree of uncertainties of X [65], and therefore having a causal effect to X [76], which is intuitively similar in a G-causality sense. The larger the value is, the greater the reduction of uncertainties [71], i.e., the higher the causality. 

As mentioned, due to the need to induce some temporal property, assume the prior discrete random variables X and Y as a Markov process (or Markov chain) with the order k and l, respectively, and n as the possible number of states. Taking one of the mentioned variables, this can be written as $X_{n}^{\left(k\right)}$ = ($X_{n-k}$ = $x_{n-k}$, …, $X_{n}=x_{n}$), and the probability distribution can be written as $P\left(X_{n+1}=x_{n+1}|\;X_{n}^{\left(n-1\right)}\right)=P\left(X_{n+1}=x_{n+1}|\;X_{n}^{\left(k\right)}\right)$.

Interestingly, this is almost identical with the G-causality’s lag time, i.e., Markov chain predicts the likelihood of the future state using only the current state (and thus is ‘memoryless’); however, it can retain some parts of its ‘past memory’ based on the given number of orders.
(8)$$\;T_{Y\to X}=-\underset{x_{n+1}}{\sum }\underset{\;x_{n}}{\sum }\underset{\;y_{n}}{\sum }P\;\left(X_{n+1=}x_{n+1},\;Y_{n}^{\left(l\right)},\;X_{n}^{\left(k\right)}\right)\;\log\frac{P\;\left(X_{n+1}=x_{n+1}|\;Y_{n}^{\left(k\right)},X_{n}^{\left(l\right)}\right)}{P\;\left(X_{n+1}=x_{n+1}|\;X_{n}^{\left(k\right)}\right)}$$

The largest distinction in Equation (8) is the inclusion of the Markov process. This will also be performed for $T_{X\;\to Y}$, and the most prominent direction of the information flow can be calculated [82] by differencing $T_{X\to Y}$ and $T_{Y\to X}$.

The measure of information flow described prior is similar to Granger’s view on the flow of causality; the notion of Granger’s G-causality with Schreiber’s transfer of ‘uncertainty’ are symmetric in this sense [65], i.e., the order of the Markov process is identical to Granger’s lag time. Transfer entropy is sometimes known as a non-linear generalization of G-causality or a form of G-causality but by looking at a more non-linear interactions with non-parametric data [83].

The next sub-section explores a more recent application of these causality measures in financial networks and the potential applications to the current uncertain climate, which is the COVID-19 pandemic.

#### 2.3.3. Causality: Effects of Financial Information Flow in International Financial Networks and the COVID-19 Pandemic

According to the cited work of [61], exploring causal networks could indicate the growth of systematic risk in the economy as well. A more practical application of information flow measures, such as transfer entropy and causal networks, can be seen practically in the works of [80]; the cited work noted that the London Stock Exchange’s technology sector, which is expressed in a causal network form, has a lower information flow, i.e., low causality.

The reasoning behind this concluded that, although the constituents are in the same sector, the companies produced different products or are large multi-sector conglomerates. Inversely, the financial services have significant outflows. This is seen similarly by the New York Stock Exchange, Tokyo Stock Exchange and the Hong Kong Stock Exchange. Other works on financial networks but with the ability to control the intensity of the causality can be seen in [61,84].

Now that information flow and its application to networks has been illustrated, the question that can be raised within the context of information flow and COVID-19 is the following; has the COVID-19 pandemic brought any information flow to Malaysia’s most lucrative market? The COVID-19 pandemic caused catastrophic impacts on the stock market worldwide, causing wide economic disruptions, investment decision delays and decreased in business profits [85]. This globally devastating impact has been reported by [86,87] as well. 

Since it is known from the cited works that COVID-19 has caused numerous devastating impacts to the world, will the constituents’ causality intensify under indices, such as the KLFIN? This is a key question as the hypothesis for this paper is that, during and after COVID-19, the causal intensity of these lucrative individual constituents will intensify regardless. Since the literature has been covered up to this point, the next section will elaborate on the methods used for this paper to prepare and process the data needed.

## 3. Methods

This section explains the data and the procedures used for processing and testing the data before further analysis is conducted on the processed data.

### 3.1. Stock Data

The KLFIN constituents’ daily closing prices dated 1 January 2018 up until the 31 December 2021, totaling 24,658 records; note from Figure 1, the total constituents for the periods selected stagnated at 30. All 30 constituents are a part of the Main Market. For analysis, the prices are grouped by year. R programming was used to process the data. Refinitiv Datastream was used as the source of data [47]. The details of these constituents can be seen on Appendix A.

Unavailable data from Refinitiv Datastream will be removed from the analysis. Constituents that have been delisted will be removed as well. Removed stock data due to non-available daily stock data from 2018 was HLGC.KL or Hong Leong Capital; delisted stock data was from TAES.KL or TA Enterprise. Thus, the final total constituent analyzed was 28. For non-trading dates, forwarding of prior data was used to avoid any null values.

As seen in Figure 3, some of the constituents either joined or left during the periods. As the leavers and joiners did not change significantly in the KLFIN, this would not be of significant issue for this paper.

### 3.2. Data Processing and Testing

Similarly to [88], a normality test was performed to ensure that the data are normally distributed; this paper used the Shapiro–Wilk normality test. However, the challenge for data processing is not necessarily normality but non-stationary data; stationary data indicate the statistical properties of that data, e.g., the mean, are stationary or constant over time, and thus past and future data have reduced changes in variation over time. This is in contrast to non-stationary data, which are bound to changes in time. 

Unfortunately, non-stationary data can lead to spurious or nonsense regressions [89] due to the effect of non-constant statistical properties. This non-stationary effect is common in stock data due to not only the timeseries but also the nature of the uncertainty of the data. A compounding issue is that standard statistical procedures are only applicable to stationary data [90]. Furthermore, it was noted that causality methods, such as G-causality, are often sensitive to non-stationary effects [69], thus, ensuring that whether the data are stationary is pertinent.

If non-stationary effects can be reduced, the data could fluctuate within a constant variation around the mean, therefore, the data can be stated to be stationary [91]. To minimize the effects of non-stationarities, the differences of the log-data between the stock dates are applied. 

The natural logarithms are also applied for some scaled properties to the data [73]. In order to check for the severity of the non-stationary effect before and after differences have been applied, unit root stationarity tests, which check the existence of a unit root (indicating the existence of non-stationarities), were performed [92]. To ensure robustness, two-unit tests were conducted: Augmented Dickey–Fuller (ADF) and Kwiatkowski–Phillips–Schmidt–Shin (KPSS).

Both of these unit tests are essentially a statistical significance test; a single unit test can be used; however, the advantage of using variations of unit tests, in this case, the ADF and KPSS, is that ADF checks period-based stationarity, i.e., differencing the timeseries, where z times could remove the unit root; KPSS, on the other hand, checks for trend stationary, where once any trends are removed (e.g., random walk), the unit root is removed [93]. 

There is however a slight difference on the basis of their null hypothesis, which is often a cause for confusion—ADF’s null hypothesis accepts the existence of a unit root, whilst KPSS’ null hypothesis accepts that trend stationarity exists. Therefore, accepting ADF’s null hypothesis indicates a unit root exists, whilst rejecting KPSS’s null hypothesis indicates the same result [94].

Once stationarity is obtained, the lag-time needs to be determined; this will be explained in the next sub-section. After the lag-time is determined, G-causality is analyzed in a form of pairwise analysis, e.g., MBBM.KL with AMMB.KL and vice versa. This is iterated throughout the pairwise within the 28 constituents; each daily closing price is grouped by year for yearly analysis. The final test was the Wald significance test to test the existence of G-causality. Once G-causality has been analyzed and the network constructed, the tested data will be used also for transfer entropy.

### 3.3. The Lag Time and Binning Criterion

#### 3.3.1. G-Causality’s Lag Time

The number of lags determines how far the prediction needs to go i.e., a lag of 1 means the model uses a single prior time to predict the future time. However, there is no solidified rule to choose the lag length. In a more Malaysian research context, annual data adopt lags of 1 or 2; quarterly data use 1 to 8 lags; and monthly data use 6, 12 or 24 lags [16].

To avoid issues with autocorrelation [95] and to ensure that the selected lag-time for each pair does not induce underfitting or overfitting, the optimal lag-time can be calculated; the Akaike Information Criterion (AIC) is used to retrieve the optimal lag-time for each of the G-causality’s pairwise combination. The AIC measures the fit of the data to a distribution; to do this, it takes the maximum-likelihood (log-likelihood), L, but induces a penalty term, k, on models with large parameter (the larger the parameter is, the more likely overfitting is induced).
(9)$$AIC=-2\ln\left(L\right)+2k\;$$

From Equation (9), a higher L indicates a higher fit to the distribution, thus, lowering the AIC value, i.e., the chosen lag-time fits the data better, therefore reducing the possibility of underfitting or overfitting. Once the optimal lag-time is calculated and retrieved, the pairwise causality is computed. Some similar pairwise analyses were conducted on [70,89,96,97]. However, the cited works performed G-causality on the indexes itself and not the individual constituents.

#### 3.3.2. Transfer Entropy’s Binning

Similarly to G-causality where the lag-time needs to be determined, transfer entropy also requires another parameter, which is the number of partitions or ‘bins’ to be considered [69]; this is needed as transfer entropy works on discrete data [98]. Furthermore, transfer entropy is somewhat sensitive to non-stationary data but may not be as sensitive as G-causality [69]; thus, verifying that stationarity is, again, important.

The number of bins can be determined by using some prior defined bins [82], equidistant binning [54], i.e., same-size bins as seen utilized in [99,100] or odd-number bins [101]. Of course, these can result in varying outcomes. For financial data, it seems reasonable to follow the works of [82,98] where quantile-based binning was utilized. With this approach, the data are binned into three separate bins based on the pre-set quantile, e.g., the 0.05–0.95 quantile; 0.025 on the extreme left side bin (i.e., extreme negative returns) and 0.025 on the extreme right side bin (i.e., extreme positive returns) and the middle bin, i.e., 0.95 containing the most recurring returns. With these parameters, volatility or non-linear behaviors may be able to be captured reasonably.

#### 3.3.3. Comparative Criteria

G-causality and transfer entropy are different approaches where one is linear and the other non-linear, respectively; a fair comparative criterion may be needed to justify the comparison between the two models. Some characteristics of somewhat similar composition between the two may be needed.

To compare the two models, this paper uses the AIC optimal lag-time on both linear and non-linear networks. Thus, the construction of the lag time (which influenced the construction of the two models) is essentially identical for both networks. Finally, the architecture of the network (sometimes known as the network type) is identical where both Granger and Schreiber’s networks are directed and weighted temporal networks.

## 4. Results

The first part of this section presents the data processing results. As mentioned previously, this paper analyzes the individual constituents pairwise combinations and not the index itself; therefore, some of the results can be large, e.g., the pairwise combinations for AIC or G-causality could lead to large combinations. Thus, several of the testing results are presented in a more concise illustration. 

Afterwards, this paper analyzes the distribution of G-causality within the KLFIN constituents. The applications of network science are used to further refine the measure by investigating the degree of the causality occurring on the network. Finally, the Schreiber transfer entropy network is constructed, and the analysis performed on G-causality is also performed on transfer entropy. All processing was conducted using the R programming language.

### 4.1. KLFIN Causality Preliminaries: Processing and G-Causality Distribution

Non-stationarities were found on the data even after the data had gone through natural logarithms. The ADF and KPSS tests confirmed that non-stationarities existed. However, the difference of 1 nullified this. The ADF and KPSS tests confirmed this afterwards; as stated from the prior section, two unit-root tests were conducted for additional validations. Interestingly, it was the KPSS test that identified the existence of non-stationarities before differencing, whilst the ADF test did not. 

Shapiro’s test confirmed the data were not significantly different from a normal distribution with a *p*-value less than 0.05. Once the tests were completed, G-causality analysis was conducted; if the Wald significance test’s *p*-value was less than 0.05, the null hypothesis was rejected, i.e., G-causality existed. 

As mentioned, the AIC will be used to determine the optimal lag-time for each pair of constitutes; thus, is it necessary to calculate the combinations for the pairs available under the KLFIN first before computing for G-Causality. Once the pairs and the AIC values are produced, G-Causality can be calculated. As stated previously, the pairwise combinations can reach some large combinations; an illustration of one of the largest sources of causality in the year 2018 is shown in Table 1. 

For the outcomes of G-causality, there are four probable outcomes [92]: no causality; two probable uni-directional causalities, e.g., from CIMB.KL to MBBM.KL and vice versa; and bi-directional causality, e.g., both CIMB.KL and MBBM.KL are causal to each other. Since it is ambiguous to look into a uni-directional causality, i.e., left–right or right–left in a statistical plot, such as a bar-plot, the total is combined.

Based on Figure 5, between the years 2018–2021, ‘No-direct’ or no causality within KLFIN’s constituents is the most prominent pattern overall; it was also the mid- and large-caps that had the highest pattern of no causality. However, it was the large-cap that had the most uni-directions (either the left to right or right to left directions) and bi-directions. This pattern was consistent for the almost half a decade of data that was used in this paper. For pre- and post-COVID-19 analysis, overall, there were no significant changes to the patterns with the exception of the sudden growth of the small-cap’s uni-directionality, as well as the growth of the large-cap’s bi- and uni-directionality post-COVID-19.

However, the source of the causality itself could not be viewed or further measured using the statistical plot alone; network science extends these measures.

In the next sub-section, the KLFIN networks are built for 2018 and 2021 for further analysis.

### 4.2. KLFIN G-Causality Market Capitalization Network

Graph theory (or in network science, a complex network) excels in finding proximities by modelling and measuring paths or directions, which is fitting for modelling and extending the measure of G-causality.
(10)$$G=\left(V,E\right)$$

A constituent in a network, G, can be represented as a node, V, and the G-causes between other node as edges, E. This is usually modelled as Equation (10) where $V=\{v_{1},\;\ldots ,\;v_{n}$}. and $E=\{e_{1},\;\ldots ,\;e_{n}$}. As mentioned in the prior sub-section, G-causality has four probable directed outcomes, and thus, a directed network is used.

A dynamic or time-varying network, sometimes also known as a temporal network would modify G over time; for this paper, a directed temporal network is used to model KLFIN G-causality. Similarly to [102]’s approach in constructing their causality network, since KLFIN data have 28 timeseries, each timeseries is associated to a node. A directed edge from a node, $v_{1}$, to another, say, $v_{2}$, indicates that $v_{1}$ G-causes $v_{2}$. 

The degree of a network node, k is the count of the edges connected to the node. The existence of causality represented as directed edges is expressed in an adjacency matrix, A. Usually, in a generic network, the out-degree (i.e., the outgoing edge from a node) and the in-degree (i.e., the incoming edge to the node) can be calculated based on Equations (11) and (12).
(11)$$k_{i}^{total}=k_{i}^{in}+k_{i}^{out}$$
(12)$$k_{i}^{total}=\underset{j}{\sum }A_{ij}+\underset{j}{\sum }A_{ji}$$

However, since the source of the causality is of interest, the out-degree is taken into account more than the in-degree. Since only the out degree is taken, only $k_{i}^{out}$ is considered, and $k_{i}^{in}$ is not taken into consideration. Conversely, Figure 5 did not consider the source causality. The measure of the source of the causality is also known as the out-degree centrality. If a particular source of causality, or focal node, is studied, this is known as ego-network analysis.

Taking all the intensity of the causality may not be helpful in analysing the causal relations relevant to a given period; much like [58,61], the intensity of the causality is analysed first. From Figure 6, during pre-COVID-19 and COVID-19, the graphs are skewed to the right, i.e., most of the causal intensity is low (the null hypothesis where G-causality does not exist is rejected with *p*-value < 0.05). 

However, notice that the positive causal intensity is already growing during COVID-19. Conversely, post-COVID-19 graphs have a completely opposite result compared to pre-COVID-19, i.e., the causal intensity is astronomically high. Since the intensity of the highest causality is high compared to the other positive causal results, seamlessly dwarfing them, this paper opted to use the highest causal value and remove the rest. Once retaining the highest causal intensity, the network was generated.

Based on Figure 7, the sources of causal effect (i.e., the out-degree) fluctuated slightly from 2018 to 2019. Regardless of the year, it seems that the large- and mid-caps were more prominent compared to the small-cap. However, it is clear that the sources of causal effect increased multiple times post-COVID-19, i.e., in 2020. The small-cap increased multiple times as well in 2020, tallying with SC’s report on small-cap growth [42].

For a micro network view, the cumulative degree of the individual KLFIN constituent is shown in Figure 8. Apart from 2020, most of the stronger causal effects came from the large- and mid-caps.

Thus far, the analysis has used G-causality, which is a linear-model; the next sub-section utilizes Schreiber transfer entropy using the same approach in network construction.

### 4.3. KLFIN Transfer Entropy Market Capitalization Network

Unlike the causal intensity derived from G-causality, the Schreiber transfer entropy has a much different distribution; this is seen in Figure 9.

Schreiber transfer entropy does not utilize the significance test; thus, values from the right scale have greater causal intensity. However, similarly to G-causality, Schreiber transfer entropy also has several weak causal intensities. For this paper, relative to the available data, weak causality for Schreiber transfer entropy is defined as values that are overly small relative to the stronger values available, e.g., 0.02 is overly small compared to 0.10 (5× smaller) and to the strongest value, which is 0.25 (almost 15× smaller). Thus, based on the available data, values lesser than 0.10 will be removed. With this threshold, some significant data are still able to be retained for further analysis.

Furthermore, as the transfer entropy causality range is not as large as G-causality, this paper also enforced another criterion to remove weak transfer entropy; when pairwise, $T_{X\;\to Y}$ and $T_{Y\;\to X}$ will be taken if the subtracted values of either pairwise is greater than 0.10. Again, even with this selected threshold, some significant data are still able to be retained for further analysis.

Identical in motivation to G-causality, the reason behind removing these weaker values is that these weaker causal values are the more prominent or dense values throughout any periods and magnified during COVID-19; thus, the weaker values will be overly protruding in any given year, engulfing the higher causal values.

Once retaining the stronger causal intensity, the network is generated. With the threshold greater or equal to 0.1, the network is formed as shown in Figure 10. Even with the removal of the more prominent weaker values, the Schreiber transfer entropy was still able to detect causal effects for the year 2018.

Similarly, to Figure 8, the cumulative degree of the individual KLFIN constituent is seen in Figure 11. Based on Figure 8 and Figure 11, it seems that both G-causality and Schreiber transfer entropy were able to capture persistent causality for the large- and mid-caps throughout the year. Interestingly, both were able to capture significant causal effects for the small-cap in 2020.

### 4.4. KLFIN Network by Bursa Malaysia’s Sub-Sector

The prior constructed and analyzed networks were based on not only the market capitalization but also using the Bursa Malaysia’s sector classification, i.e., Financial Services. For the latter, however, the classification ’Financial Services’ is generic; thus, it limits the paper’s ability to analyze the specifics of a financial constituent causality (e.g., the constituent’s business activity with other constituents).

To further analyze individual constituents’ business activities and the impact of causal networks on the understanding of these activities, this paper uses Bursa Malaysia’s sub-sector classifications, which consist of Banking, Insurance and Other Financials. For the details of this, refer to [103]. Based on these sub-sectors, the prior networks are regenerated and shown in Figure 12 and Figure 13 for G-causality and Schreiber transfer entropy, respectively.

Based on Figure 12, it is clear that the Insurance sub-sector persisted and grew larger post-COVID-19, shrinking but maintaining its visibility in 2021. It is interesting that Banks and Other Financials followed this trend as the year progressed.

Notice in Figure 13, although the year 2018’s causality is far weaker compared to other subsequent years, it is interesting to note that two of the nodes came from the Other Financials sub-sector and none from the Insurance sub-sector. However, the Insurance sub-sector became increasingly prominent as the year progressed.

## 5. Discussion

The focal point of this paper is identifying and analyzing the existence of causality using G-causality and Schreiber transfer entropy within the KLFIN’s 28 constituents. This study differs from other studies [70,89,96,97] where the cited works performed G-causality on the indexes itself and not the individual constituents. This was demonstrated during pre- and post-COVID-19, which can be considered as one of the most uncertain times in the modern world. The data used were also a year before, during and after COVID-19, and thus this paper was able to observe the uncertainty impacting the KLFIN during the selected periods. 

The results were visualized and analyzed using not only classical statistical descriptive analysis but also using directed temporal networks where the nodes were weighted by the out-degree centrality. The constructed directed temporal networks were not only successful in capturing sources of causal extremes but also even indicated that both networks overlapped in results even though both differed in their implementation as seen in Equations (1) and (8). 

As the networks were able to detect causal extremes even in the KLFIN, which is known to be the strongest sector in the Main Market, the hypothesis that the causal intensity of these lucrative individual constituents will intensify regardless of uncertain times is accepted.

The statistical distribution for G-causality indicates that a high number of the KLFIN constituents did not have any causal effects from 2018 up until 2021 (illustrated in Figure 5). Uni-directional causality analysis indicates that the large and small-caps grew throughout post-COVID-19. For bi-directional causality analysis, the large-cap had the highest bi-directional causality, followed by the mid-cap, and the small-cap seems to be non-existent with the exception of the year 2020. 

However, this statistical distribution alone would not be able to take into consideration the fluctuating degree of the source of the causality; hence, the paper turns to measures in network science—in particular, the out-degree centrality depicted in Equations (11) and (12). The comparison between the two financial networks was performed, not only on the basis of their identical network architecture, i.e., both networks are directed and weighted temporal networks but also on the basis that their lag-time was set to be identical with each other (see Section 3.3.3). Both weighted temporal networks also took in only the stronger causal intensity by thresholding only the higher causal values (see Section 4.2 and Section 4.3).

Based on Equations (1) and (8), although it is clear that G-causality and transfer entropy have differing equations but overlapping concepts (illustrated in Section 2.3.2), both constructed financial networks captured identical patterns of growing out-degrees between pre- and post-COVID-19. However, before thresholding these weighted networks, the growth of causality before and after COVID-19 had high amounts of weak causal values (illustrated in Figure 6 and Figure 9), therefore, engulfing the more intense causal values. 

Naturally, if these higher number of weak causalities are not removed or processed, the network will become too dense, and thus the more prominent causality would not be identified effectively. Nevertheless, even after removing these weaker values and retaining only the stronger causal values, the two networks were still able to capture the constituents’ causalities, simultaneously portraying a far stronger and relevant causal effect on the constructed networks. These results can be seen in Figure 7, Figure 10, Figure 12 and Figure 13.

Furthermore, even after the weaker causality was removed from both networks, it is interesting to observe that both G-causality and transfer entropy were able to detect identically increasing out-degree distributions for the small-cap in 2020, tallying with the RHB and the SC’s reports on the small-cap growth on 2020 (shown in Figure 7 and Figure 10, respectively). Both networks also identified similar out-degree distribution where the large- and mid-caps were the dominant sources of causal effects on the KLFIN (illustrated in Figure 7 and Figure 10). Related to the topic of patterns, both networks were also able to detect the fluctuating causal pattern of the Shariah-compliant constituents where the causal source appeared to fluctuate as the year progressed. This can be seen in Figure 12 and Figure 13.

Finally, also seen in Figure 12 and Figure 13 and with the weaker causality removed from the networks, the Insurance sub-sector network was still the highest source of increasing causality, persistently identified by both networks. However, from the same Figures, the transfer entropy network was able to detect far more causalities from the Insurance sub-sector nodes, whilst the G-causality network detected far fewer.

The latter network also produces a declining causal intensity from the Insurance sub-sector nodes. For the same Figures but focusing on the year 2020 only, conversely to the G-causality network, the transfer entropy network was able to uniquely identify extremely intense causal effect from two ego networks—namely, Apex Equity Holdings Berhad (APES.KL) and Pacific & Orient Berhad (PACO.KL). 

Performing some essential fundamental analysis on these two constituents, APES.KL is a small-cap constituent principally engaged in the business of investment holding and trading in marketable securities [104], whilst PACO.KL is a mid-cap investment holding constituent engaged in providing financial services and information technology services [105]. Conversely, however, both G-causality and Schreiber transfer entropy networks were able to detect one particular ego’s identical pattern of highly-intense causal effect during the same period as well; this is the MAA group or MAAS.KL; a small-cap investment based constituent involved in general and life insurance [106] (Figure 7 and Figure 10). 

These three major sources of causality may have been the most influential constituents in Malaysia during the COVID-19 pandemic. What is interesting from these three constituents as well is that they are either small- or mid-cap sized constituents but are sources of highly-intense source of causality. Note that the unexpected highly intense causal impacts from these smaller-cap constituents during uncertain and devastating times tallies with the growth reports from the SC and RHB mentioned on Section 2.1.1.

The two prior cited works were not only generated using different methods but were also analyzed and produced by these two completely different sources. Although the cited works came from different sources, the cited authors’ results are consistent with the results of this paper, i.e., the higher impact of the smaller caps against the large-cap during COVID-19; thus, further analysis to bridge the results from these different sources with this paper may be conducted in the future. This type of discovery might be also visible on other indices, which researchers can revisit in the future.

Related to the topic of different sources, the Central Bank of Malaysia or Bank Negara Malaysia (BNM), which is responsible for the financial system stability of the country [107], recently produced the Financial Sector Blueprint 2022–2026 strategic plan [108] to ensure that the country moves forward after the aftermath of the pandemic. One of the key reforms due to the aftermath of the pandemic is digitalization of the financial sector; as customers are now more adaptive with digital and remote access due to the impact of the movement controls and general health concerns. Digitalization is one the key reforms focused by BNM. 

Interestingly, BNM cited financial institutions increased usage of not only artificial-intelligence and machine-learning technologies but cloud computing as well. This may have led some of the financial sectors (in particular, those that are active in technology services, such as PACO.KL) to have their causality increased through the pandemic. 

As well, BNM cited that, due to the pandemic, a greater appreciation and demand for insurance was realized. It seems this finding by the BNM tallies with that of the findings of the networks produced in this paper. Since the Blueprint attempts to improve various insurance policies and offerings, the work performed by this paper may also support the perceived value of the insurance constituent’s influential power to other constituents.

Nevertheless, other interesting perspectives from these new findings, in particular the sheer size of the intensity of the causality can also be induced; a source of high-intense causality may be seen as a source of influential power or perception due to its high quantities of causal effects on the effected constituent. However, the source of this high-intense causality itself may not be seen as a source of persistent stability (as investing in the larger cap is not mutually inclusive to stability, as discussed in Section 2.1.1). 

This may be inferred from Figure 12 and Figure 13 where both G-causality and Schreiber transfer entropy networks did not detect high-intense causalities on the Shariah-compliant stock constituents; however, in the year 2021, a sudden and abnormal surge of high-causality was seen in one of the Shariah-compliant stocks—BIMB.KL or Bank Islam Malaysia, a banking sub-sector constituent. However, during the uncertain and damaging period of 2019, none of the Shariah-compliant stock constituents were amongst the highest in both networks. 

Although the next enquiry would be the reason behind this abrupt phenomenon occurring and the investigation behind the conventional constituents’ stable high causal effect, this paper acknowledges that the sample size of the Shariah-compliant stock constituents is rather small and the results for this specific analysis are still on a preliminary level but, nevertheless, they suggest a potential future research work. On a final note, this paper also acknowledges limitations in terms of the stationarity estimations and finite size effects and analysis performed in this paper [109] (see Section 3.2). 

From a high-level view, the difference in the outcomes between G-causality and Schreiber transfer entropy are possibly due to the non-linear nature of transfer entropy and its ability to detect outliers [109,110]. If the financial market is viewed as a complex system, non-linearity [109,111] and outliers in view of power laws [112] are to be expected. 

Nevertheless, with these concepts and cited works in mind, linear and non-linear measures can mathematically be demonstrated to obtain varying results as well [113]. Finally, it is also beyond the scope of this paper to analyze the root cause of the causality as this requires further fundamental analysis on each constituent’s business ventures. Furthermore, other lag times, thresholding criteria, binning methods and even the current quantile-based binning method can also be further investigated and refined. 

Extending this to the smaller LEAP and ACE markets (discussed on Section 2.1.2) could uncover more insights to the markets as well. Other future investigations and measures, such as performing comparative analysis on the effectiveness of using mutual information and correlation on the constituents, as well as analyzing the robustness of either the conventional or Shariah-compliant constituents can be the subjects of future research [114,115].

## 6. Conclusions

The COVID-19 pandemic has caused severely devastating impacts and uncertainties in the world. Nevertheless, identifying a constituent’s causality regardless of stature, specifically for those companies that are listed under the highly lucrative KLFIN, during such devastating times may provide not only relief but also an added novel approach toward investment decision making. This paper’s findings showed that both networks were not only successful in capturing sources of causal extremes but also overlapped in their results even though both differed in their implementation. 

Unexpectedly as well, both networks showed that the small- and mid-caps had high sources of causal intensity during and after COVID-19. As both networks were able to detect causal extremes even in the KLFIN’s lucrative constituents, the hypothesis that the intensity of these lucrative individual constituents will intensify regardless of uncertain times is accepted. 

Though conducted separately by distinct institutions, other cited works (such as the SC’s report) prominent banks (such as RHB and even the Central Bank of Malaysia) showed supportive findings for this research, and there were similarities in the findings of small- and mid-cap growth regarding the need to digitalize the financial sectors and the impacts on financial services providers. The supportive findings from these distinct institutions reaffirmed the findings of this paper’s work and affirmed the rich prospects for future works as well.

## Figures and Tables

**Figure 1 entropy-24-01100-f001:**
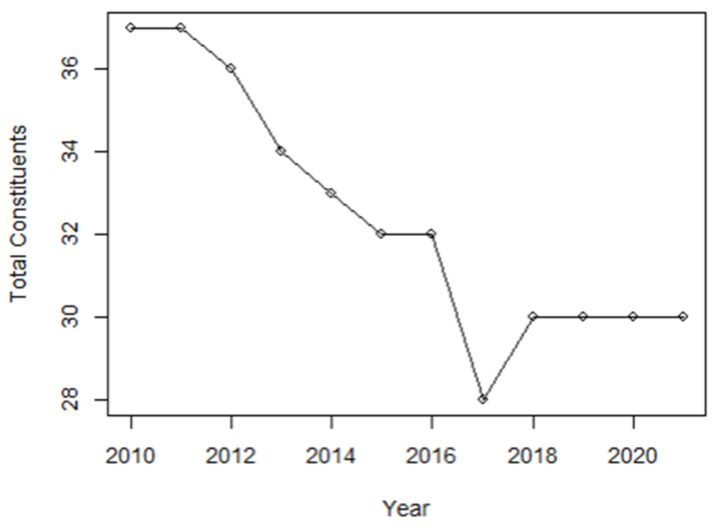
Leavers and joiners of the KLFIN over the decade up until 2021 [47].

**Figure 2 entropy-24-01100-f002:**
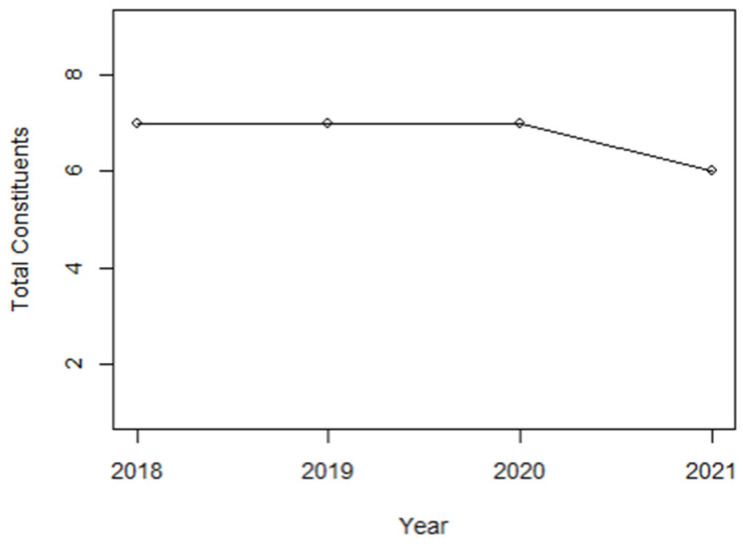
Leavers and joiners of the KLFIN constituents listed in KLCI between 2018 and 2021 [47].

**Figure 3 entropy-24-01100-f003:**
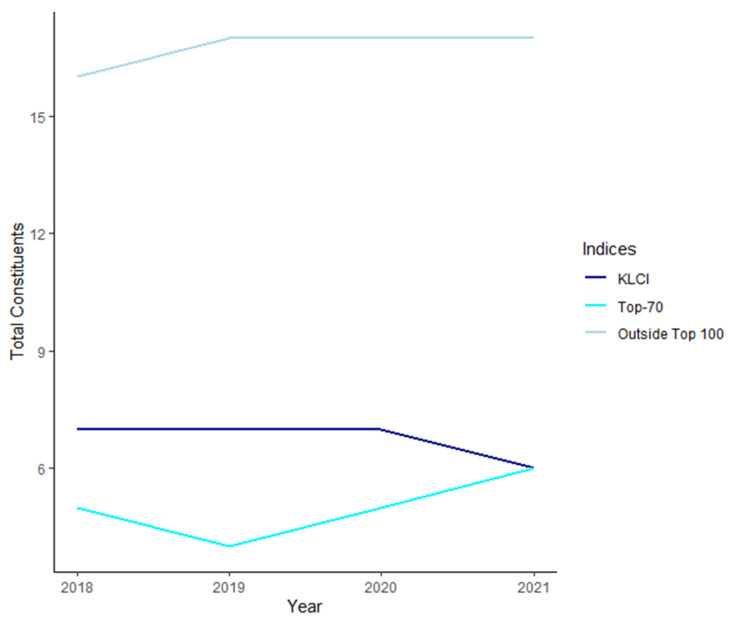
Leavers and joiners of the KLFIN constituents within and outside of the Top 100 [47].

**Figure 4 entropy-24-01100-f004:**
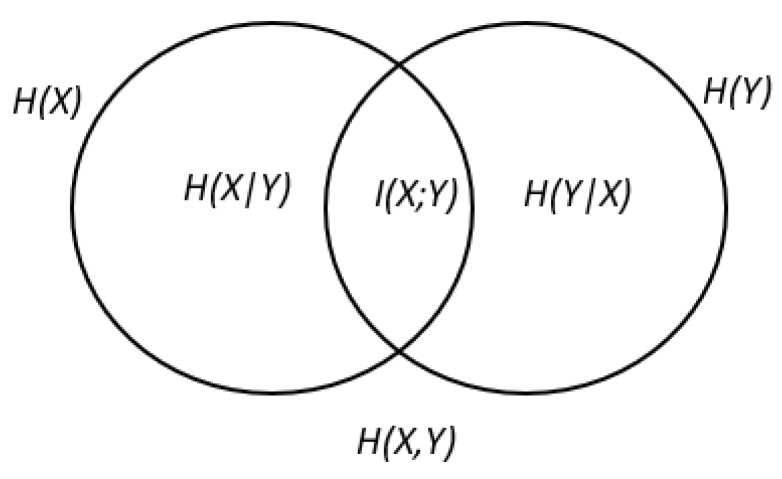
Venn diagram illustrating Equations (2)–(7).

**Figure 5 entropy-24-01100-f005:**
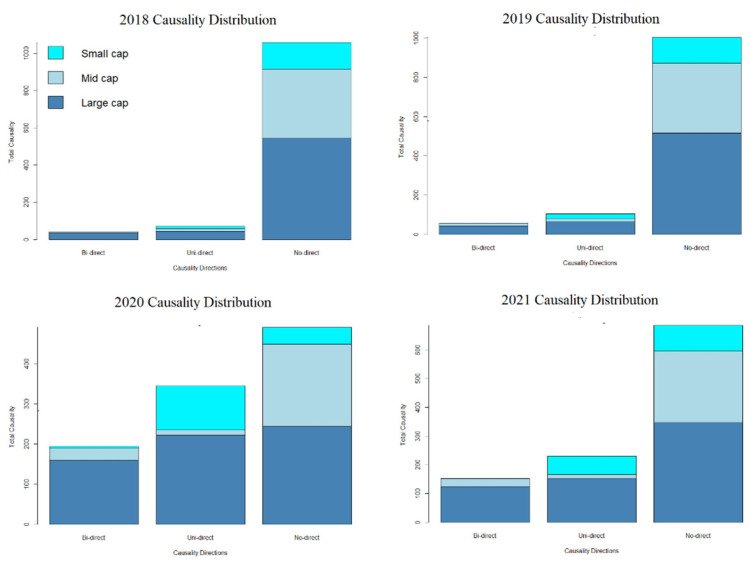
KLFIN G-causality distribution from 2018 to 2021.

**Figure 6 entropy-24-01100-f006:**
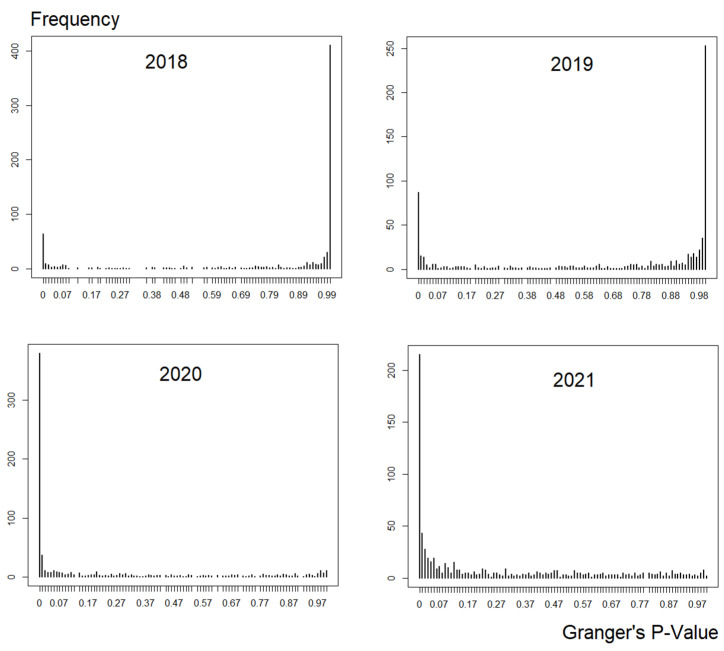
Granger’s frequency of the strength of the causality from 2018 to 2021.

**Figure 7 entropy-24-01100-f007:**
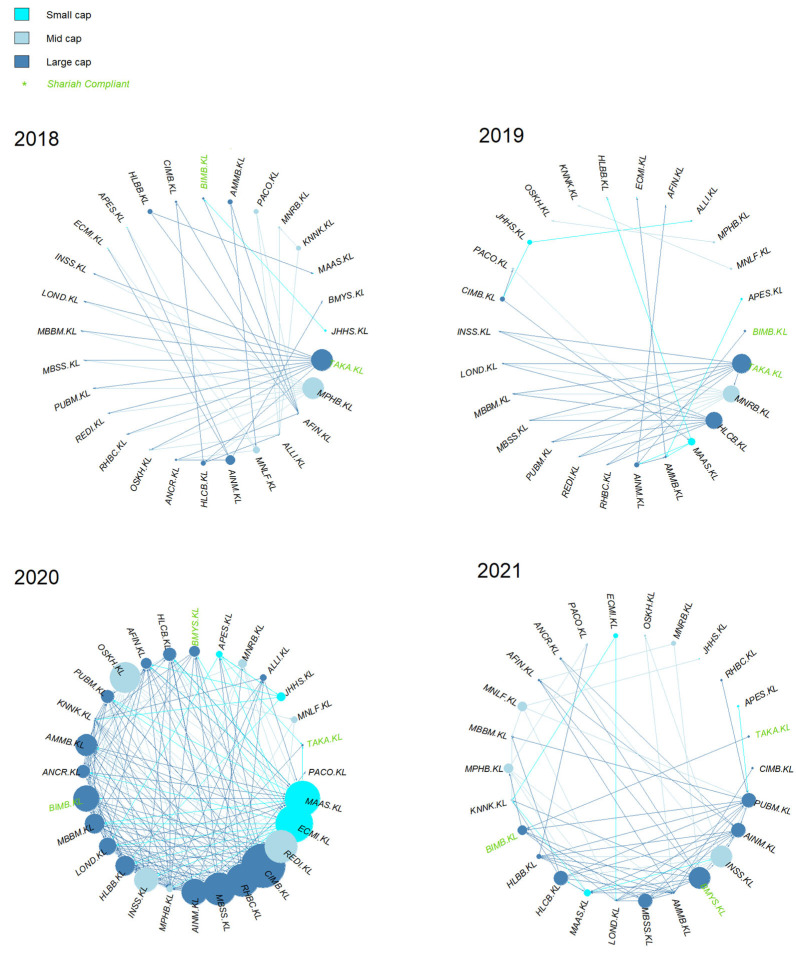
Overall KLFIN G-causality network formation from 2018 to 2021.

**Figure 8 entropy-24-01100-f008:**
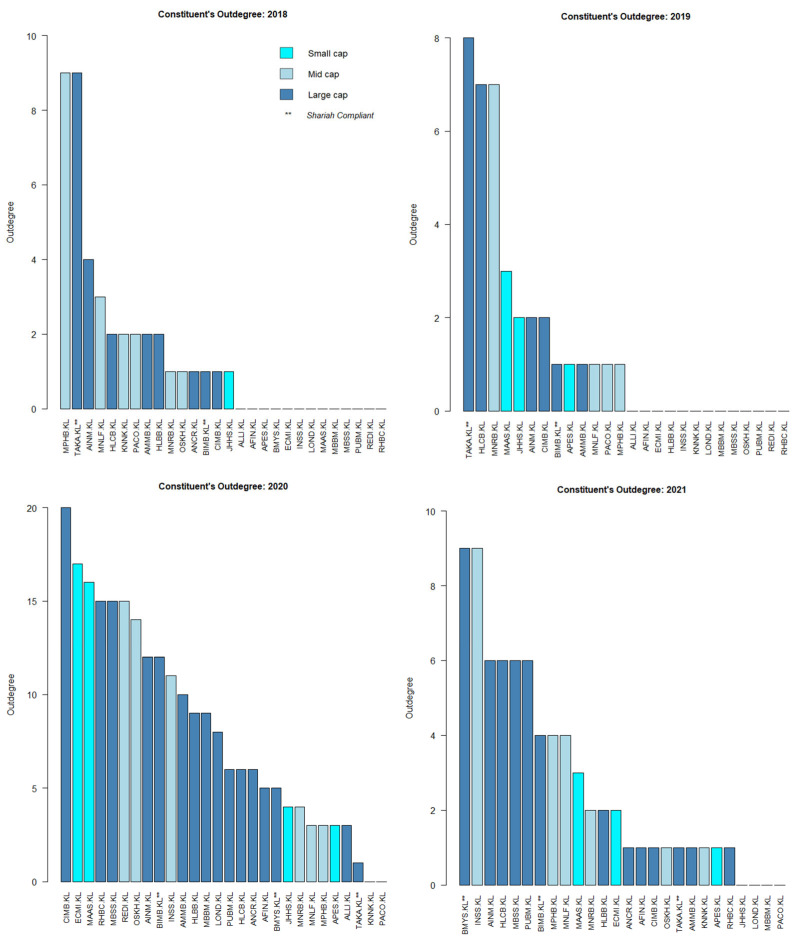
KLFIN G-causality degree and hub centrality from 2018 to 2021.

**Figure 9 entropy-24-01100-f009:**
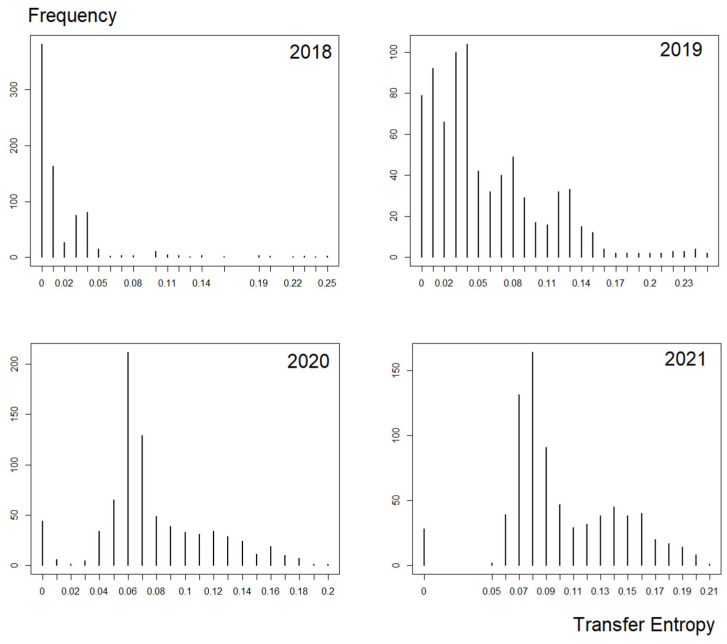
The transfer entropy’s frequency of the strength of the causality from 2018 to 2021.

**Figure 10 entropy-24-01100-f010:**
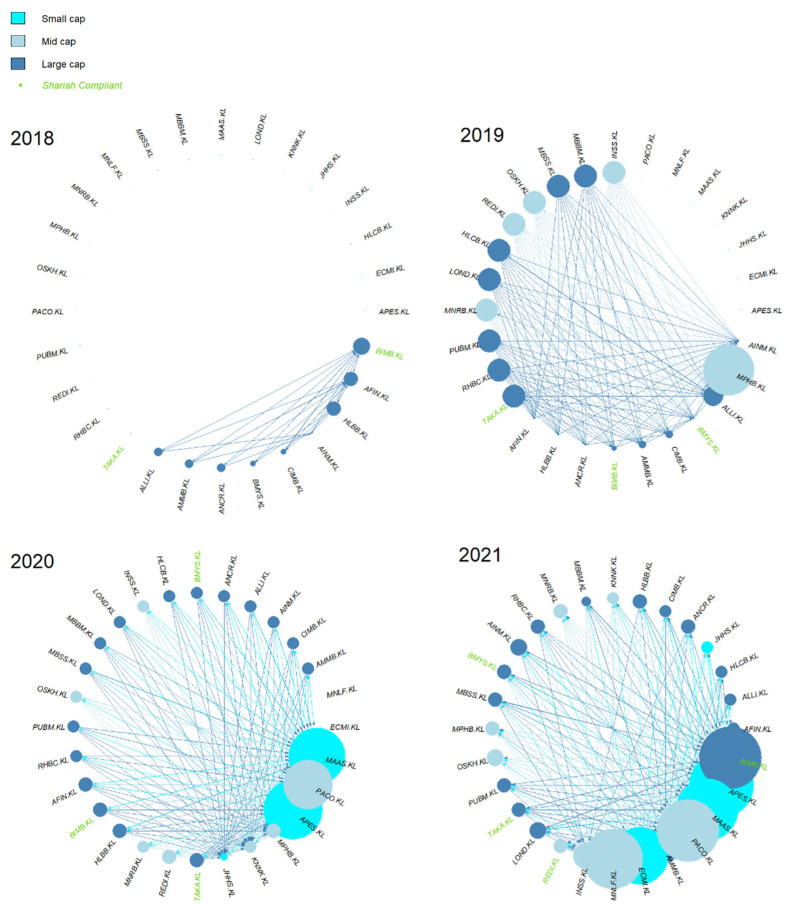
Overall KLFIN transfer entropy network formation from 2018 to 2021.

**Figure 11 entropy-24-01100-f011:**
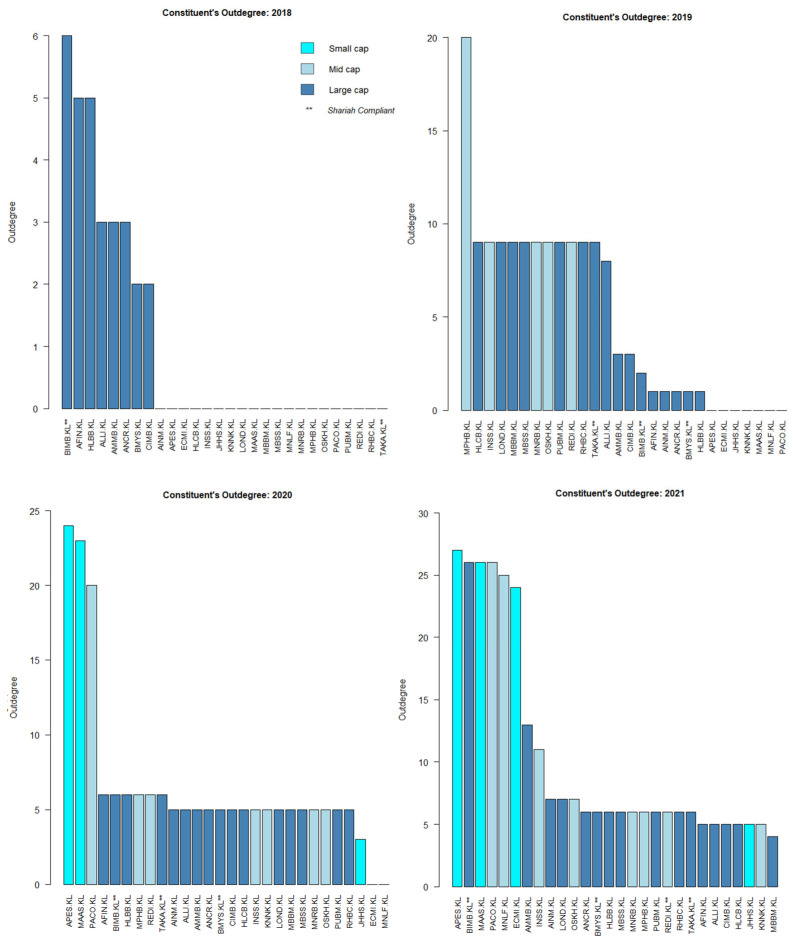
KLFIN transfer entropy degree and hub centrality from 2018 to 2021.

**Figure 12 entropy-24-01100-f012:**
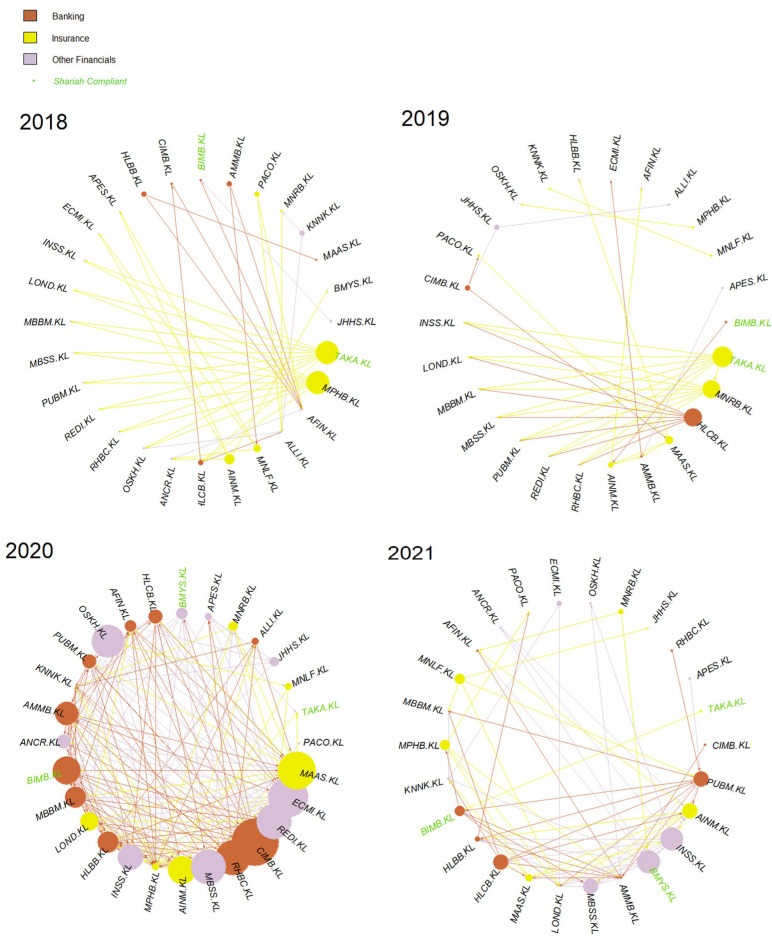
KLFIN Granger network sub-sector.

**Figure 13 entropy-24-01100-f013:**
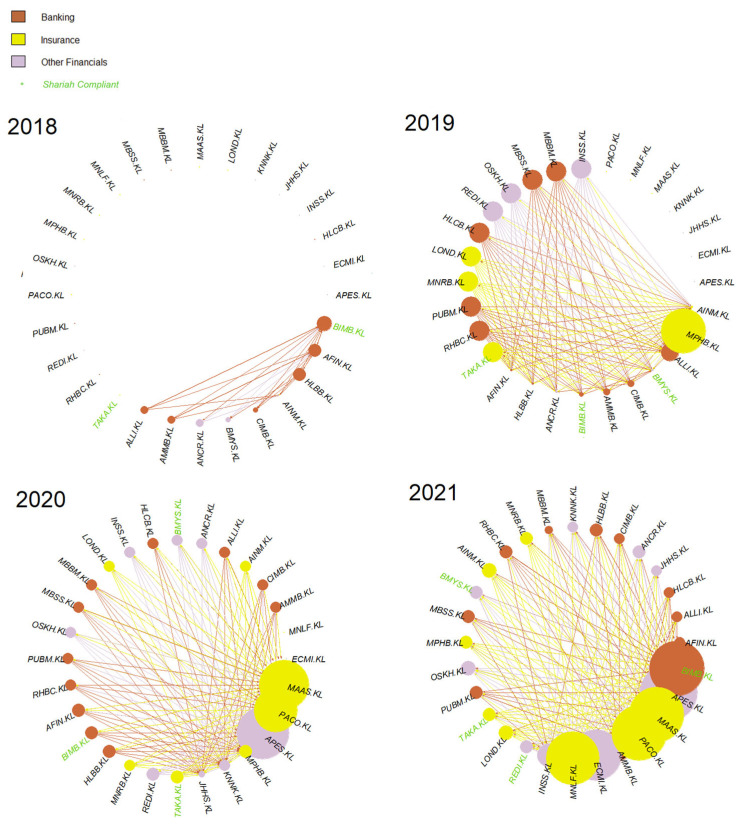
KLFIN Schreiber transfer entropy network sub-sector.

**Table 1 entropy-24-01100-t001:** Pairwise combination results of one of the highest sources of causality in 2018—MPHB.KL.

Constituent 1	Constituent 2	AIC	Wald-Significance’s *p*-Value	G-Causality
MPHB.KL	ALLI.KL	1	0.01278	Yes
MPHB.KL	AFIN.KL	7	0.00105	Yes
MPHB.KL	AMMB.KL	1	0.02037	Yes
MPHB.KL	HLBB.KL	8	0.00002	Yes
MPHB.KL	HLCB.KL	1	0.00000	Yes
MPHB.KL	INSS.KL	1	0.00000	Yes
MPHB.KL	LOND.KL	1	0.00000	Yes
MPHB.KL	MBBM.KL	1	0.00000	Yes
MPHB.KL	MBSS.KL	1	0.00000	Yes
MPHB.KL	OSKH.KL	1	0.00000	Yes
MPHB.KL	PUBML.KL	1	0.00000	Yes
MPHB.KL	REDI.KL	1	0.00000	Yes
MPHB.KL	RHBC.KL	1	0.00000	Yes

## Data Availability

All relevant stock data retrieved from Refinitiv Datastream.

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
