# Peer review of "The Causality and Uncertainty of the COVID-19 Pandemic to Bursa Malaysia Financial Services Index’s Constituents"

_entropy, 2022, doi:10.3390/e24081100_

Round 1

Reviewer 1 Report

General Comments

From my point of view, the paper is fairly well written, and it tackles an interesting and timely topic. The authors used a Granger causality and Schreiber transfer entropy for analyzes the Bursa Malaysia’s KLFIN index constituents’ causal performance from 1st of January 2018 (pre-COVID-19) up until the 31st of December 2021 (post-COVID-19). The study shows that that both Granger causality and Schreiber transfer entropy were able to detect patterns of increasing causality from pre- to post-COVID-19 but with differing intensities of detected causality. Furthermore, the Insurance sub-sector is one of the index’s largest sources of causality.  The paper consists of following sections: Introduction, Bursa Malaysia’s Large, Mid and Small-Cap: Performance Comparison Pre- and Post-COVID-19, Bursa Malaysia Financial Services Index: Perception and Evolution, Causality, Stock Data, Processing and Testing, Analysis, Discussions.

However, I find some recommendations:

1.        The abstract must contain the main purpose of the paper.

2.        The "Introduction" should briefly place the study in a broad context.

3.        I consider that the introduction should specify the novelty of the paper compared to other papers published in this area.

4.        It would be very useful to add in the "Introduction" section the purpose, objectives and hypothesis of the research.

5.        Though the authors have included a few recent studies on the topic, I recommend the authors to refer to other recent works indexed in Web of Science, Scopus, Emerald, Cambrige, and of course MDPI Journals. We suggest that the authors cite papers published in MDPI journals, such as:

5.1) Ştefan Cristian Gherghina, Daniel Ştefan Armeanu, Camelia Cătălina Joldes, 2020, Stock market reactions to Covid-19 pandemic outbreak: Quantitative evidence from ARDL bounds tests and Granger causality analysis, Int. J. Environ. Res. Public Health, 17, 6729; doi:10.3390/ijerph17186729

5.2) Elena Razumovskaia, Larisa Yuzvovich, Elena Kniazeva, Mikhail Klimenko, Valeriy Shelyakin, 2020, The effectiveness of Russian government policy to support smes in the COVID-19 pandemic, J. Open Innov. Technol. Mark. Complex., 6, 160; doi:10.3390/joitmc6040160

5.3) Esam Mahdi, Víctor Leiva, Saed Mara’Beh, Carlos Martin-Barreiro, 2021, A new approach to predicting cryptocurrency returns based on the gold prices with support vector machines during the COVID-19 pandemic using sensor-related data, Sensors, 21, 6319; https://doi.org/10.3390/s21186319

5.4) Muhammad Azmat Hayat, Huma Ghulam, Maryam Batool, Muhammad Zahid Naeem, Abdullah Ejaz, Cristi Spulbar, Ramona Birau, 2021, Investigating the Causal Linkages among Inflation, Interest Rate, and Economic Growth in Pakistan under the Influence of COVID-19 Pandemic: A Wavelet Transformation Approach, J. Risk Financial Manag., 14, 277; https://doi.org/10.3390/jrfm14060277

5.5) Mariana Hatmanu, Cristina Cautisanu, 2021, The impact of COVID-19 pandemic on stock market: evidence from Romania, Int. J. Environ. Res. Public Health, 18, 9315; https://doi.org/10.3390/ijerph18179315

5.6) Lilko Dospatliev, Miroslava Ivanova, Milen Varbanov, 2022, Effects of COVID-19 Pandemic on the Bulgarian Stock Market Returns, Axioms, 11, 94; https://doi.org/10.3390/axioms11030094

5.7) Thomas Chinan Chiang, 2022, Evidence of economic policy uncertainty and COVID-19 pandemic on global stock returns, J. Risk Financial Manag., 15, 28; https://doi.org/10.3390/jrfm15010028

5.8) Xiuping Ji, Sujuan Wang, Honggen Xiao, Naipeng Bu, Xiaonan Lin, 2022, Contagion Effect of Financial Markets in Crisis: An Analysis Based on the DCC–MGARCH Model, Mathematics, 10, 1819; https://doi.org/10.3390/math10111819. 

6.        The "Causality" section is good, and the research methodology used in the study is appropriate for a study of this nature.

7.        Authors have also explained well various tests are being used to test the data.

8.        However, authors have not presented some of the test statistics such as Shapiro-Wilk normality test, ADF unit test, KPSS unit test and Wald significance test.

9.        In the sub-sections of 5.3 there is a wrong numbering. It should be 5.3.1, 5.3.2 and 5.3.3.

10.    Authors have discussed the results well but they haven't also compared the findings of this study with that of previous studies.

11.    The "Conclusion" section is missing. Nevertheless, the paper has to some extent expressed its case, measured against the technical language of the field and the expected knowledge of the journal's readership.

12.    I propose to the authors the following structure of the paper: Introduction; Literature review; Methods and Results; Discussions and Conclusions

Reviewer 2 Report

Dear Author,

I am always direct in my nature of the communication of review reports. Well done. The paper is simple but impactful. However, the execution can be improved. We will now discuss that. I like the simple way of writing the work, this will be very easy to read to the general readership. In this regard, I am giving it a minor revision and propose the following changes in numeric order, so that it is easier for you to address them.

1.       Articles discussing current context can give you broader scope in your literature review as we are going through the covid time. You can add them to your discussion. I would strongly suggest adding recent last 3 years paper discussing both COVID and GFC in different context including European Banking sector or safe Heaven in well established journals like Financne research letter, Annals of Operations Research or Global Finance journal. This will strengthen your argument on the fact that you are integrating the same ideology. Also, no reference that is less than 5 years old. You are working on a contemporary perspective stay on that.

3. I would like to see the regulatory implication at the conclusion.

4. The intro part should be divided into three parts: Why this is important, what are we doing, and what are our findings.

5. The article should be corrected for typos.

Best of luck. Keep up the good work.
